# The Role of Human Chorionic Gonadotropin Beta (hCGβ) in HPV-Positive and HPV-Negative Oropharyngeal Squamous Cell Carcinoma

**DOI:** 10.3390/cancers14122830

**Published:** 2022-06-08

**Authors:** Anni Sjöblom, Timo Carpén, Ulf-Håkan Stenman, Lauri Jouhi, Caj Haglund, Stina Syrjänen, Petri Mattila, Antti Mäkitie, Jaana Hagström

**Affiliations:** 1Department of Pathology, University of Helsinki and Helsinki University Hospital, P.O. Box 21, FI-00014 Helsinki, Finland; timo.carpen@fimnet.fi (T.C.); jaana.hagstrom@hus.fi (J.H.); 2Department of Otorhinolaryngology—Head and Neck Surgery, University of Helsinki and Helsinki University Hospital, P.O. Box 263, FI-00029 HUS Helsinki, Finland; lauri.jouhi@helsinki.fi (L.J.); petri.mattila@hus.fi (P.M.); antti.makitie@helsinki.fi (A.M.); 3Research Program in Systems Oncology, Faculty of Medicine, University of Helsinki, P.O. Box 63, FI-00014 Helsinki, Finland; 4Department of Clinical Chemistry, University of Helsinki and Helsinki University Hospital, P.O. Box 63, FI-00014 Helsinki, Finland; ulf-hakan.stenman@pp.fimnet.fi (U.-H.S.); stisyr@utu.fi (S.S.); 5Research Programs Unit, Translational Cancer Medicine, University of Helsinki, P.O. Box 63, FI-00014 Helsinki, Finland; caj.haglund@helsinki.fi; 6Department of Surgery, University of Helsinki and Helsinki University Hospital, P.O. Box 440, FI-00029 Helsinki, Finland; 7Department of Pathology, Turku University Hospital, Kiinamyllynkatu 10, 20520 Turku, Finland; 8Division of Ear, Nose and Throat Diseases, Department of Clinical Sciences, Intervention and Technology, Karolinska Institutet and Karolinska Hospital, SE-171 76 Stockholm, Sweden; 9Department of Oral Pathology and Oral Radiology, University of Turku, Lemminkäisenkatu 2, 20520 Turku, Finland

**Keywords:** OPSCC, hCGβ, HPV

## Abstract

**Simple Summary:**

Oropharyngeal squamous cell carcinoma (OPSCC) is often associated with human papillomavirus (HPV). The management of OPSCC remains a challenge, and HPV-negative disease is commonly associated with impaired survival. Novel insights into diagnostics and treatment modalities are warranted to improve disease outcome. Various potential biomarkers affecting prognosis in OPSCC have been distinguished. Our aim was to study the role of the free β-subunit of human chorionic gonadotropin hormone in HPV-negative and HPV-positive OPSCC.

**Abstract:**

Background: This study was carried out to observe the upregulation of the free β-subunit of human chorionic gonadotropin (hCGβ) and its prognostic significance in human papillomavirus (HPV)-positive and HPV-negative oropharyngeal squamous cell carcinoma (OPSCC). Materials and methods: A total of 90 patients with OPSCC treated with curative intent at the Helsinki University Hospital (HUS), Helsinki, Finland, during 2012–2016 were included. Serum samples were collected prospectively, and their hCGβ concentrations (S-hCGβ) were determined by an immunofluorometric assay. The expression of hCGβ in tumor tissues was defined by immunohistochemistry (IHC). HPV determination was performed by combining p16-INK4 IHC and HPV DNA PCR genotyping. Overall survival (OS) and disease-specific survival (DSS) were used as survival endpoints. Results: S-hCGβ positivity correlated with poor OS in the whole patient cohort (*p* < 0.001) and in patients with HPV-negative OPSCC (*p* < 0.001). A significant correlation was seen between S-hCGβ and poor DSS in the whole cohort (*p* < 0.001) and in patients with HPV-negative OPSCC (*p* = 0.007). In a multivariable analysis, S-hCGβ was associated with poor DSS. Of the clinical characteristics, higher cancer stage and grade were associated with S-hCGβ positivity. No statistically significant correlation with tissue positivity of hCGβ was seen in these analyses. Conclusion: S-hCGβ may be a potential independent factor indicating poor prognosis, notably in HPV-negative OPSCC.

## 1. Introduction

The incidence of oropharyngeal squamous cell carcinoma (OPSCC) appears to be increasing in developed countries [1,2,3]. The majority of new OPSCC cases are associated with human papillomavirus (HPV) infection [4]. HPV is recognized as one of the most important risk factors in OPSCC, together with smoking and alcohol use [5]. The prognosis of HPV-negative OPSCC is adverse; 5-year survival is about 50% among patients who are HPV-negative and 80% among patients who are HPV-positive [6]. HPV-positive OPSCC is widely agreed upon as representing the distinct disease entity compared with HPV-negative OPSCC [7]. The role of HPV in the disease pathogenesis and survival has not been thoroughly studied. Common treatment modalities include surgery with or without adjuvant (chemo-)radiotherapy, or only (chemo-)radiotherapy [8]. Even with successful treatment, a patient’s quality of life often declines afterwards. The development of novel methods for diagnostics and treatment is thus highly desirable.

Novel prognostic factors, such as various serum and tissue specific biomarkers, may be valuable for developing management and for improving outcome assessment. Individualized treatment plans could be generated based on pre-treatment biomarker levels, similar to the development of methods for determining HPV status [9]. A potential prognostic biomarker is the free β-subunit of the human chorionic gonadotropin (hCGβ) [10,11].

Human chorionic gonadotropin (hCG) is a heterodimeric glycoprotein secreted mainly by trophoblastic cells in the placenta [11]. Different forms of hCG can be observed in subjects who are pregnant or not pregnant. The upregulation of hCGβ is often seen in non-trophoblastic malignancies [12,13], including cancer of the breast [14], liver, and pancreas [15,16]. The upregulation of hCGβ in cancer is commonly linked to impaired prognosis [13,15,16]. Additionally, there is evidence of increased hCGβ concentrations and expression in both serum and in tumor tissue in OPSCC as well as in other cancers of the head and neck [17,18,19]. However, as far as we are aware, the upregulation of hCGβ has not been previously studied while comparing HPV-positive and HPV-negative OPSCC.

The aim of our study was to investigate the serum concentrations and tissue expression of hCGβ and their correlation with clinical characteristics and prognosis in HPV-positive and HPV-negative OPSCC.

## 2. Materials and Methods

### 2.1. Study Population

The study includes 90 patients with OPSCC from an existing database described in previous studies [20,21]. Patients with prospectively collected serum samples and established HPV statuses were admitted to the analysis. HPV status was based on both p16-INK4 immunohistochemistry (IHC) and PCR with Luminex-based multiplex HPV DNA genotyping, as defined in our previous work [20]. The patients were treated with a curative intent at the Helsinki University Hospital (HUS) during 2012–2016, either by surgery with or without postoperative (chemo)radiotherapy, or by definitive (chemo)radiotherapy. Details of the clinical characteristics, treatment, and survival were collected from the electronic patient archives. The characteristics included patient age at diagnosis, gender, cigarette smoking, heavy alcohol use, TNM class (8th edition), stage, differentiation grade, cancer localization, and HPV status of the OPSCC sample. These characteristics were compared to the serum concentration of hCGβ (S-hCGβ) and to the hCGβ tissue expression. In addition, we compared the S-hCGβ levels to the serum concentrations of tumor-associated trypsin inhibitor (S-TATI). The methodology of the S-TATI assay is described in our previous work [21].

The Research Ethics Board at the HUS approved the study design, and an institutional permission was granted (Dnr: 51/13/03/02/2013). Written consent of approval was acquired from all participating patients.

### 2.2. S-hCGβ Determination

S-hCGβ was determined by an in-house time-resolved immunofluorometric assay, as described previously [22]. The limit of detection was 0.5 pmol/L, and the coefficient of variation (CV) was <10% at concentrations of 5–5000 pmol/L. The upper reference limit for S-hCGβ in the serum was 2 pmol/L. Age or gender did not have a significant effect on the reference range. The selected cut-off value for our analysis was 2 pmol/L, based on the 97.5th percentile for apparently healthy subjects. The method as well as the upper-reference limit were validated previously [22,23,24].

### 2.3. Tumor Slides

To observe a possible correlation between the tissue expressions of hCGβ and S-hCGβ, tumor slides cut from paraffin blocks with thicknesses of 3.5 µm were prepared for immunohistochemistry (IHC). The tumor material from all patients whose S-hCGβ exceeded 1.5 pmol/L resulted in 9 slides for immunostaining. Additionally, 9 S-hCGβ-negative patients (S-hCGβ < 0.9 pmol/L) were selected for the IHC as controls. This resulted in a total of 18 tumor slides for the analysis. Two samples of placental tissue served as positive controls.

### 2.4. hCGβ Immunohistochemistry

The protocol for immunostaining is described in detail in our previous work [21]. We used a monoclonal anti-hCGβ antibody (MAb 9C11) produced in-house [25] as a primary antibody. This antibody is specific for hCGβ and shows no reaction with intact hCG [26].

The samples were immunoscored by two researchers (A.S. and J.H.). The hCGβ positivity was evaluated according to the percentage of stained nuclei; a sample was considered positive if >10% of the nuclei were stained. The intensity of the staining, which was mainly high, was not taken into consideration. The immunostaining of hCGβ is demonstrated in Figure 1.

### 2.5. Data Analysis

The statistical analyses were performed with IBM SPSS software version 27. The χ^2^-test and the Fischer’s exact test were performed for the crosstab comparisons between hCGβ and the clinical characteristics. The independent samples *t*-test was used for the normally distributed continuous variables. For the survival analysis, the endpoints were overall survival (OS) and disease-specific survival (DSS), signifying the time periods from the first day of diagnosis until the end of the follow-up, or the time of death of any cause or from the disease, respectively. Survival was assessed with a log-rank test and illustrated with the Kaplan–Meier method. The Cox regression analysis was used for the univariable and multivariable survival analyses. Variables resulting in *p*-values below 0.05 in the univariable analysis were subjected to a multivariable analysis. To achieve a normal distribution for S-hCGβ, the values were logarithmically transformed. *p*-values below 0.05 were considered statistically significant.

## 3. Results

### 3.1. Clinical Characteristics and S-hCGβ

The median for S-hCGβ was 0.50 pmol/L (range 0.10–6.70 pmol/L). In this cohort, 7 (7.8%) of the patients were S-hCGβ-positive (cut-off value 2 pmol/L) and 83 (92.2%) were S-hCGβ-negative. The majority (*n* = 4/7) of the S-hCGβ-positive patients were HPV-negative, yet this finding was not statistically significant. A strong correlation was seen between S-hCGβ positivity and higher stage (*p* = 0.013) as well as between S-hCGβ positivity and higher tumor grade (*p* = 0.039). No correlation was found between S-hCGβ and other clinical characteristics (Table 1).

Additionally, we observed a correlation (*p* = 0.028) between S-hCGβ positivity and S-TATI positivity. However, we found no correlation between the tissue immunostaining of TATI and hCGβ.

### 3.2. Clinical Characteristics and hCGβ IHC

Of the tumor slides available for analysis, 12 patients (66.7%) were hCGβ tissue-positive and 6 (33.3%) were hCGβ tissue-negative. The hCGβ IHC analysis revealed no significant association between tissue expression and clinical characteristics. and tissue positivity of hCGβ showed no correlation with S-hCGβ (Supplemental Appendix A).

### 3.3. Survival and S-hCGβ

In the Kaplan–Meier analysis, S-hCGβ positivity was strongly associated with poor OS in all patients (*p* < 0.001) and in the subgroup of patients who were HPV-negative (*p* < 0.001). In addition, we found a significant correlation between S-hCGβ and poor DSS in the whole cohort (*p* < 0.001) and in HPV-negative patients (*p* = 0.007). The survival curves are presented in Figure 2.

In the multivariate Cox regression analysis, poorer OS was significantly linked with age (adjusted hazard ratio (HR) 1.08, 95% confidence interval (CI) 1.03–1.14, *p* = 0.003) and smoking (*p* = 0.024). Poorer DSS was significantly correlated with S-hCGβ (HR 2.31, 95% CI 1.20–4.45, *p* = 0.012) and age (HR 1.08, 95% CI 1.02–1.14, *p* = 0.013). Assessed individually, S-hCGβ showed significant correlations with adverse OS (*p* = 0.001) and DSS (*p* < 0.001). The results of the multivariate analysis are presented in Table 2.

## 4. Discussion

There are currently no prognostic serum markers incorporated in the diagnostics or the treatment process of head and neck cancers. Based on our current findings, the determination of S-hCGβ levels prior to treatment may offer novel information about prognosis and facilitate the selection of optimal treatment modalities. Thus, a prospective determination of S-hCGβ may have value in the management OPSCC. Interestingly, several similarities can be seen between our results and the findings of previous studies concerning hCGβ upregulation in other cancers [15,23,24,26]. However, our small sample size limits robust conclusions, and similar studies with larger cohorts are required to validate the present findings. 

To the best of our knowledge, the present study is the first one to offer novel information on the prognostic significance of hCGβ in HPV-negative and HPV-positive OPSCC. By applying a multivariable model in the statistical analysis, we were able to identify the independent prognostic potential of S-hCGβ. Findings similar to ours have been observed earlier, although in these studies, HPV status was not taken into consideration and, furthermore, the sample sizes were small [17,18]. Additionally, our study is the first to utilize both serum and tumor tissue in patients with OPSCC. However, benign conditions associated with hCGβ upregulation were not taken into consideration in our analysis [27,28].

Various mechanisms may mediate cancer-related hCGβ regulation, but these mechanisms remain to be thoroughly explained [16]. The ability of cancer cells to secrete hCGβ has been speculated on previously [11], and hCGβ has been shown to inhibit apoptosis in gynecological cancers [29,30]. Both mechanisms described above may be responsible for promoting tumorigenesis in OPSCC, and our findings support this hypothesis. Furthermore, as a majority of the patients in our IHC analysis showed hCGβ tissue positivity, we assume that hCGβ is synthesized in the OPSCC cells.

Our results showed a correlation between S-hCGβ and higher stages and grades in OPSCC. Similar results have been presented in ovarian and gastric cancer [23,24]. Our results showed no correlation between S-hCGβ and tissue expression, and similar findings have been seen in renal cancer [26]. Various studies have reported on the function of hCGβ as an autocrine growth-promoting factor in cancer [23,31,32]. As larger tumors constitute higher stages in cancer patients and higher stages are a strong prognostic factor for impaired survival [33,34], our findings of the correlation between elevated S-hCGβ and higher stage align with these studies. The mechanism of the growth-promoting capabilities of hCGβ is not known, but it has been speculated that hCGβ may disrupt the actions of various growth factors with growth-inhibiting characteristics, such as transforming growth factor-β (TGF-β) [35]. Furthermore, cell membrane’s phospholipid methylation induced by hCGβ upregulation has been reported previously [17], and this function promotes tumor growth through different mechanisms [36,37,38].

In previous studies on prostate cancer, the elevated expression of TATI has been reported during the endocrine treatment of prostate cancer [39]. In this study, the elevation of TATI promoted epithelial–mesenchymal transition, which was associated with impaired prognosis. Similar findings have been observed in ovarian cancer associated with the increased expression of hCGβ [40]. There are significant differences in the methodologies between these studies, and they may not be comparable with that of our study. However, as stated above (Table 1), in our results, the serum positivity of hCGβ correlated with the serum positivity of TATI, and we speculate that the upregulation of hCGβ and TATI may be associated with epithelial–mesenchymal transition in OPSCC as well.

Based on our current results, HPV status may have a moderate effect on hCGβ levels. However, the size of our study population is not large enough to draw definitive conclusions. Recently, it was shown that HPV16 E7 induced promoter-methylation and decreased CGB3 gene expression, which may cause the absence or lower levels of hCGβ-immunostaining in HPV-positive tumors [41]. Interestingly, the Cox regression analysis showed a significant independent correlation between reduced DSS and S-hCGβ (Table 2.). Furthermore, we observed a weak correlation between shorter OS and S-hCGβ, although this result was not statistically significant (*p* = 0.060). However, our results show definitive proof that S-hCGβ has independent prognostic value irrespective of HPV status.

It is possible that the tissue expression of hCGβ may be substantial despite low intracellular content, as hCGβ is not stored in cells but secreted instantly following synthesis [42]. We speculate that this phenomenon may partially explain the possible discrepancies in our analysis. However, the sample size of our IHC analysis was small (n = 18) and it is not possible to reach definitive conclusions based on our findings. The material for this analysis was scarce, as a majority of the patients received definitive (C)RT without surgery, in which case, the only tissue material available was a small biopsy. 

## 5. Conclusions

According to our findings, increased serum concentrations of hCGβ may be an independent marker of poor prognosis in OPSCC, especially among patients with HPV-negative OPSCC. We are advocating for further prospective research with larger study populations to validate these findings.

## Figures and Tables

**Figure 1 cancers-14-02830-f001:**
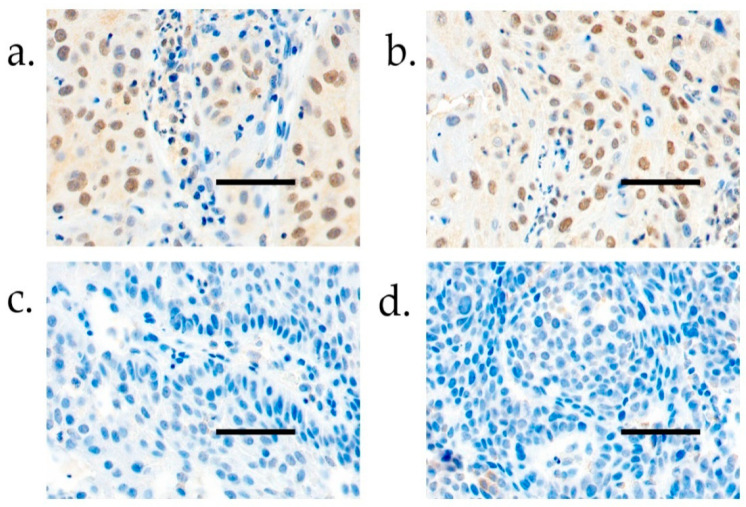
(**a**) Positive hCGβ immunoexpression in tumor tissue. (**b**) Positive hCGβ immunoexpression in tumor tissue. (**c**) Negative hCGβ immunoexpression in tumor tissue. (**d**) Negative hCGβ immunoexpression in tumor tissue. Scale bar length, 100 µm. Magnification, ×400.

**Figure 2 cancers-14-02830-f002:**
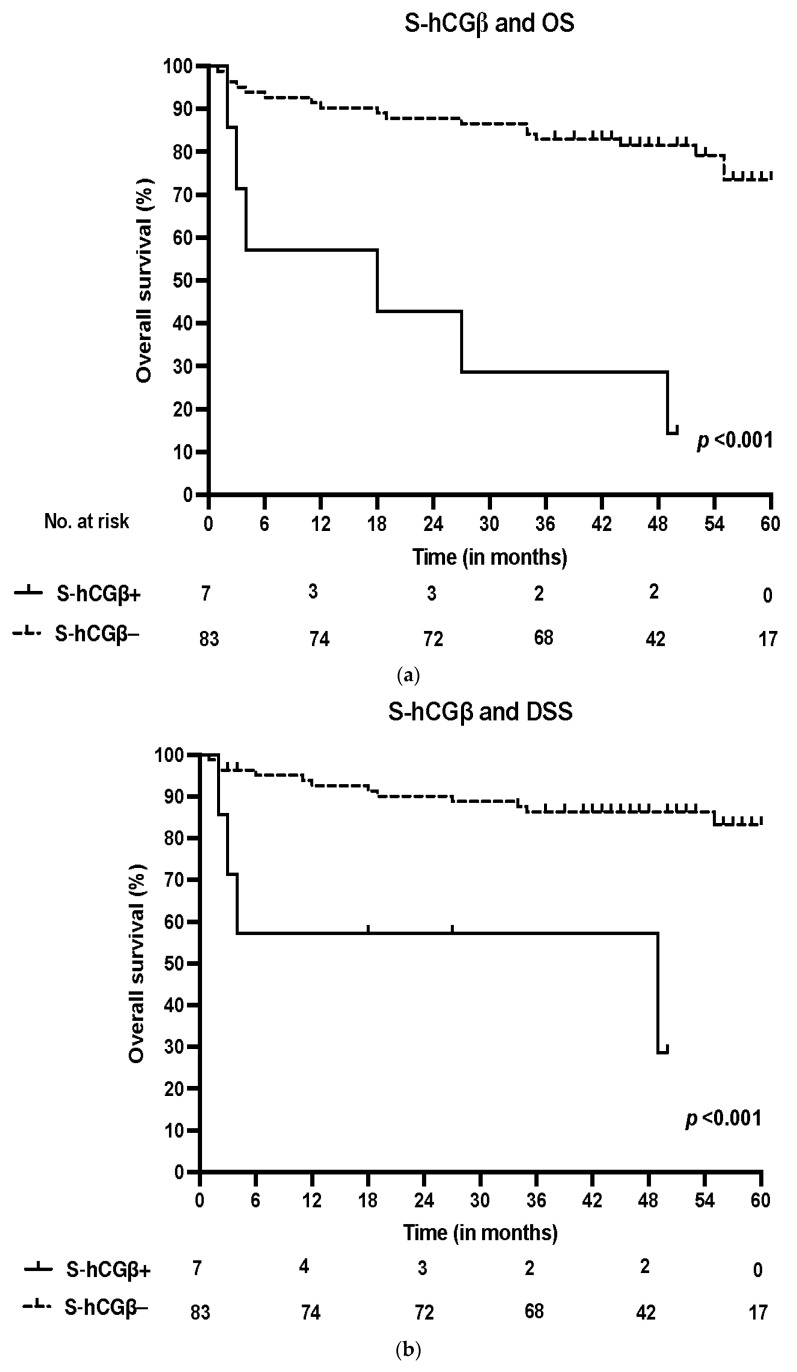
(**a**) Overall survival (OS) according to positive (>2 pmol/L) and negative (<2 pmol/L) S-hCGβ in the whole patient cohort. (**b**) Disease-specific survival (DSS) according to positive and negative S-hCGβ in the whole patient cohort.

**Table 1 cancers-14-02830-t001:** Clinicopathological data according to positive (>2 pmol/L) and negative (<2 pmol/L) serum concentrations of hCGβ (S-hCGβ).

Variable.	S-hCGβ+	%	S-hCGβ−	%	*p*-Value	Missing/%
Number of patients	7	7.8	83	92.2		
Mean age at diagnosis	64.5		61.4		0.156	
Gender						
Male	5	70.6	61	92.4		
Female	2	8.3	22	91.7	0.906	
Smoking						
Non-smoker	1	3.6	27	96.4		
Ex-smoker	2	6.7	28	93.3		
Current smoker	4	12.5	28	87.5	0.554	
Heavy alcohol use						
Non-drinker	2	4.7	41	95.3		
Ex-drinker	1	10.0	9	90.0		
Current drinker	3	14.3	18	85.7	0.383	16/17.8
T class						
T1–T2	3	5.2	55	94.8		
T3–T4	4	12.5	28	87.5	0.214	
N class						
N0–N1	5	6.8	69	93.2		
N2–N3	2	12.5	14	87.5	0.437	
Stage						
I–II	2	3.2	61	96.8		
III–IV	5	18.5	22	81.5	0.013 *	
Grade						
I	0	0.0	3	100.0		
II	4	26.7	11	73.3		
III	3	4.2	69	95.8	0.039 *	
Localization						
Tonsil	4	7.5	49	92.5		
Base of tongue	1	4.5	21	95.5		
Soft palate	0	0.0	10	100.0		
Posterior wall of oropharynx	2	40.0	3	60.0	0.097	
HPV						
HPV+	2	3.8	51	96.2		
HPV−	5	13.5	32	86.5	0.090	
S-TATI						
S-TATI+	4	19.0	17	81.0		
S-TATI-	3	4.3	66	95.7	0.028 *	

Abbreviations: hCGβ: Human chorionic gonadotropin β; HPV: Human papillomavirus; S-TATI: Serum concentration of tumor-associated trypsin inhibitor. *p* < 0.05 *.

**Table 2 cancers-14-02830-t002:** Multivariate Cox regression analysis for overall survival (OS) and disease-specific survival (DSS).

Variable	OS			DSS		
	HR	95% CI	*p*-Value	HR	95% CI	*p*-Value
Age at diagnosis	1.08	1.03–1.14	0.003 *	1.08	1.02–1.14	0.015 *
Smoking			0.024 *			0.183
Ex-smoker versus never smoker	1.28	0.30–5.52	0.738	0.55	0.09–3.34	0.513
Current smoker versus never smoker	4.56	1.35–15.43	0.015 *	2.45	0.64–9.37	0.192
Stages III–IV versus Stages I–II	1.67	0.68–4.10	0.267	1.93	0.60–6.20	0.267
HPV− versus HPV+	1.13	0.40–3.22	0.818	1.02	0.26–3.93	0.980
Ln(S-hCGβ)	1.71	0.98–2.98	0.060	2.31	1.20–4.45	0.012 *

HR: Hazard ratio; CI: Confidence interval. S-hCGβ values are log-transformed (Ln). *p* < 0.05 *.

## Data Availability

The data presented in this study are available from the corresponding author upon request. The data are not publicly available due to patient data security.

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
