# Peer review of "The Role of Human Chorionic Gonadotropin Beta (hCGβ) in HPV-Positive and HPV-Negative Oropharyngeal Squamous Cell Carcinoma"

_cancers, 2022, doi:10.3390/cancers14122830_

Round 1

Reviewer 1 Report

The authors describe the prognostic role of serum and tissue S-hCGß in HPV positive and negative oropharyngeal squamous cell carcinoma. 90 patients were included in the evaluation and 83% were S-hCGß negative. 7 patients showed positivity for S-hCGß in serum and tissue. The positivity correlated with poor overall survival especially in HPV-negative patients.

Questions to the authors:
- tissue analysis of S-hCGß was only available in 18 patients? If so, please comment on that in the discussion section.

The results have to be interpreted with more care regarding the small sample size. Otherwise the manuscript is well written and needs only minor modifications.

Author Response

RESPONSE TO THE REVIEWER #1 COMMENTS:

Comment 1: Tissue analysis of S-hCGß was only available in 18 patients? If so, please comment on that in the discussion section. The results have to be interpreted with more care regarding the small sample size. Otherwise the manuscript is well written and needs only minor modifications.

Response 1: We would like to thank the Reviewer #1 for these valuable comments and the contribution. Our sample size is indeed small, however, as similar results concerning the serum concentrations of hCGß have been seen in other cancers cited in our manuscript, we believe our results are promising (Please see refrences 15, 23, 24 and 26, lines 300 – 301, 315 – 317 and 320 - 321). The material for the tissue analysis was scarce, as majority of the patients received definitive (C)RT without surgery, in which case the only tissue material available was a small biopsy. We acknowledge the small amount of samples, and to highlight this, we have mentioned in the manuscript that we saw no significant correlation in the IHC analysis (Please see lines 45 – 46, 156 – 158 and Supplemental table 1. lines 355 -364). The suggested specifications about the sample size have now been added in the discussion section in the revised version of the manuscript (Please see lines 191 – 193 and 243 -246).

END OF AUTHOR’S RESPONSE TO REVIEWER #1

Reviewer 2 Report

Major
1. Authors  have very low number of positive cases, which are 7/90. It will be appropriate to do a continuous analysis using cox proportional hazards model and then testing the correlation of hCGb with OS or DFS. To the very least, does hCGb expression correlate with survival in those 7 patients?

2. The study seems to be underpowered because most of the conclusions are made based on the characteristics of 7 patients. Authors have used the reference levels for hCGb from healthy individuals which may not represent the distribution of hCGb expression in OSCC patients.

3. Do you have a control/ reference protein that remains more or less constant, assayed along with hCGb in the serum of these patients. The levels of reference gene will control for the technical variables. You can also normalize the levels of hCGb with your reference protein to be more accurate.

4. The IHC cohort doesnot resemble the distribution of the serum cohort in terms of number and hCGb positivity, hence making meaningful conclusions is difficult.

Minor

Line 75-76 is incomplete.

Full form of CV on the line 101

Author Response

RESPONSE TO THE REVIEWER #2 COMMENTS:

Comment 1: Authors have very low number of positive cases, which are 7/90. It will be appropriate to do a continuous analysis using cox proportional hazards model and then testing the correlation of hCGb with OS or DFS. To the very least, does hCGb expression correlate with survival in those 7 patients?

Response 1: We thank the Reviewer #2 for this note and we appreciate the contribution. We have performed both univariable and multivariable Cox regression analyses using OS and DSS as our end-points, and serum hCGß as a continuous variable (Please see lines 133 – 141 and 165 – 170, and Table 2. lines 176 - 178). The whole patient cohort (n=90) was included in this analysis as presented in Table 2.

Comment 2: The study seems to be underpowered because most of the conclusions are made based on the characteristics of 7 patients. Authors have used the reference levels for hCGb from healthy individuals which may not represent the distribution of hCGb expression in OSCC patients.

Response 2: We thank the Reviewer #2 for this comment. We recognize that the number of patients is rather small, but it is still larger than any other study on this subject. Furthermore, the multivariable survival analysis in our study was performed with the whole cohort (n=90) and in the analysis serum hCGß showed clinical prognostic significance. Hence, we believe our study provides new information on the potential role of serum hCGß in HPV-positive and HPV-negative OPSCC. We have addressed this limitation further in the revised version of the manuscript (Please see lines 191 - 193)

As a decision limit, we used the upper reference limit for serum hCGß in healthy subjects. This is a common method to select cut-off levels for disease markers, including tumor markers. Similar reference limits have been used in previous studies cited in our manuscript (Plese see references 23, 24, 26, lines 315 – 317 and 320 - 321).

Comment 3: Do you have a control/ reference protein that remains more or less constant, assayed along with hCGb in the serum of these patients. The levels of reference gene will control for the technical variables. You can also normalize the levels of hCGb with your reference protein to be more accurate.

Response 3: We thank the Reviwer #2 for this comment. To our knowledge, reference proteins are not commonly used in tumor marker studies, and the rationale behind this is not clear. There are no recommendations on how to select reference proteins in studies like ours.

Comment 4: The IHC cohort doesnot resemble the distribution of the serum cohort in terms of number and hCGb positivity, hence making meaningful conclusions is difficult.

Response 4: The Reviewer #2 is making a valuable point and we agree. The material for the IHC analysis was scarce, as majority of the patients received definitive (C)RT (without surgery), in which case the only tissue material available was a small biopsy. This matter has been acknowledged and discussed in the revised version of the manuscript (Please see lines 243 -246). However, as mentioned above, the patient whole patient cohort was included to the survival analysis concerning serum hCGß.

Comment 5: Line 75-76 is incomplete.

Response 5: We thank the Reviewer #2 for this note. The error has been corrected in the revised version of the manuscript (Please see lines 75 - 76)

Comment 6: Full form of CV on the line 101

Response 6: We thank the Reviewer #2 for this note. The full form has been added in the revised version of the manuscript (Please see lines 101 - 102)

END OF AUTHOR’S RESPONSE TO REVIEWERS

Round 2

Reviewer 2 Report

The authors have satisfactorily addressed the concerns and I approve the manuscript in the current form.